

# Effects of strength training on repeated sprint ability in team sports players: a systematic review

Andrés Osses-Rivera[1], Rodrigo Yáñez-Sepúlveda[2], Sebastián Jannas-Vela[3], Jeppe F. Vigh-Larsen[4] and Matías Monsalves-Álvarez[5]

[1] Escuela de Educación, Universidad Mayor, Santiago, Chile
[2] Faculty Education and Social Sciences, Universidad Andrés Bello, Viña del Mar, Chile
[3] Instituto de Ciencias de la Salud, Universidad de O'Higgins, Rancagua, Chile
[4] Department of Sports Science and Clinical Biomechanics, University of Southern Denmark, Odense, Denmark
[5] Exercise and Rehabilitation Sciences Institute, Faculty of Rehabilitation Sciences, Universidad Andrés Bello, Santiago, Chile

Corresponding author
Matías Monsalves-Álvarez,
matias.monsalves@unab.cl

## ABSTRACT

**Objective.** This systematic review was conducted to analyze the existing evidence on the effects of strength training (ST) and complex/contrast training (CCT) on repeated sprint ability (RSA) in team sports players.

**Methods.** A systematic review of the literature was performed following the PRISMA statement. PubMed, Web of Science, and Scopus databases were used. Original full-text articles were analyzed, without date restriction until May 26, 2024, written in English, peer-reviewed, and for eligibility must have included (1) male or female team sports players, amateur or professional category, without age restriction (2) lower extremity ST and/or CCT program (3) active control group (4) running RSA test (*e.g.*, repeated shuttle sprint ability test or straight-line repeated sprint ability test) before and after the intervention period (5) controlled trial.

**Results.** A total of 3,376 studies were identified and screened. Finally, 10 articles were included based on the inclusion and exclusion criteria, all with moderate methodological quality according to the PEDro scale. The best time, mean time, and total time presented significant pre and post-test changes, using ST in 3, 2, and 1 experimental groups, respectively, and using CCT in 1, 1, and 1 experimental groups, respectively, with almost no differences in the percentage decrement most commonly reported in RSA tests. There were no changes in the control groups.

**Conclusion.** Together, ST performed in a range of maximal power provides benefits in the best time and mean time and performed between 80 to 95% of 1 repetition maximum (RM) provides benefits in the best time, mean time, and total time in RSA tests. CCT performed between 75 to 90% of 1 RM combined with jumps and sprints provides benefits in the best time, mean time, and total time in RSA test, but no unaltered percentage decrement in ST and CCT in elite and semi-professional team sport players.

## INTRODUCTION

Team sports are characterized by an intermittent high-intensity exercise pattern, including sequences of repeated maximal or near maximal efforts ≤10s interspersed with short recovery intervals ≤60s (*Spencer et al., 2005*; *Girard, Mendez-Villanueva & Bishop, 2011*). During a match, the frequency, duration and distance of sprinting on average range 7–61 of 2.0 s for 15.2 m, 18–105 of 0.5–2.4 s for 3.9–9.5 m, 19-25 of 0.9–3.0 s for 10.0–19.1 m and 7.4 ± 3.3 of 1.6–1.9 s for 7.8–13.0 m, in elite male soccer, basketball, handball, and futsal players respectively (*Taylor et al., 2017*).

The ability to maintain sprint performance as maximal efforts are repeated has been termed repeated sprint ability (RSA) (*Spencer et al., 2005*; *Girard, Mendez-Villanueva & Bishop, 2011*), and even though recent evidence has questioned the frequency by which such sequences occur (*Schimpchen et al., 2016*; *Taylor et al., 2016*), they may indeed represent critical "worst case" scenarios during games where fatigue has been shown to manifest both transiently after intense efforts, as well as during the latter stages of a match (*Mohr, Krustrup & Bangsbo, 2005*). For example, in soccer, the RSA tests have been reported to be valid in discriminating playing levels and highly reliable (*Altmann et al., 2019*).

In general, the physiological factors of importance for team sport performance are derived from the initial sprint performance and the ability to sustain performance during subsequent bouts, that is, maximal and repeated sprint ability (*Bishop & Girard, 2013*). Ultimately, sprint performance is determined by the stride length and frequency, where stride length is related to muscular power, elastic properties of musculotendinous unit/tissue, and dynamic flexibility (*Bishop, Girard & Mendez-Villanueva, 2011*; *Bishop & Girard, 2013*). In contrast, stride frequency is related to intramuscular coordination (*Bishop, Girard & Mendez-Villanueva, 2011*; *Bishop & Girard, 2013*), which includes neuromuscular activation (*i.e.,* recruitment of motor units (MU) and the discharge rates of action potentials, MU synchronization, and neuromuscular inhibition) (*Girard, Mendez-Villanueva & Bishop, 2011*). Single sprint performance is predominantly fueled by phosphocreatine hydrolysis and anaerobic glycolysis (*e.g.,* almost 50/50 during a single 6-s sprint) (*Gaitanos et al., 1993*; *Parolin et al., 1999*). At the same time, the importance of aerobic metabolism increases with repeated sprint efforts (*Gaitanos et al., 1993*; *Parolin et al., 1999*). Recovery between sprints is strongly related to phosphocreatine resynthesis capacity, ultimately determined by aerobic capacity (*Bishop, Girard & Mendez-Villanueva, 2011*; *Bishop & Girard, 2013*). In contrast, other factors, such as hydrogen buffering and restoration of ionic homeostasis, may also be important (*Sahlin & Harris, 2011*; *Hostrup & Bangsbo, 2017*). As such, multiple factors interact to determine repeated sprint ability, with strength/power and metabolic factors being key components.

Importantly, greater muscular strength is strongly associated with enhanced force-time characteristics (rate of force development (RFD) and external mechanical power), with both being critical muscular factors that determine sprint performance in team sports (*Suchomel, Nimphius & Stone, 2016*) and the evidence is compelling that sprint performance can be enhanced by different modalities of strength training programs (*Suchomel, Nimphius & Stone, 2016*).

Traditional strength training (ST) is characterized by high loads (70–100% 1 repetition maximun (1 RM)) performed with low repetitions, typically between 12-1 and separated by longer durations of passive rest of approximately 2–5 min. Instead, power training is characterized by using loads >80% 1 RM for enhanced strength characteristics or 30–60% 1RM for enhanced velocity characteristics, 3-6 sets, 1-6 reps, and >3 min rest (heavy) or 1–2 min rest (moderate) respectively (*Kraemer & Ratamess, 2004*). In contrast, complex/contrast training (CCT) is characterized by a combination of high-load, low-velocity strength exercise, also called conditioning activity (CA), followed immediately by a low-load, high-velocity plyometric/ballistic exercise, set by set within the same session (*Cormier et al., 2022*). This sequence elicits the "post-activation potentiation" effect with a short half-life of (∼28 s), classically defined as an improvement in muscle strength that results from phosphorylation of myosin regulatory light chain by myosin light chain kinase in the muscle fibers type II (*Blazevich & Babault, 2019*).

In the last decade, a considerable amount of published studies as meta-analyses have analyzed the effects of ST (characterized by actions against external resistance) and CCT of the lower extremities, highlighting substantial improvements in sprint performance in team sports players (*Seitz et al., 2014*; *Freitas et al., 2017*; *Thapa et al., 2020*). On the other hand, the effects of ST and CCT on RSA are less clear because, unlike a single sprint, initial sprint performance is associated with a more significant subsequent drop in performance. It is unclear how a potential strength-mediated improvement in sprint performance affects repeated sprint performance when several bouts are performed (*Girard, Mendez-Villanueva & Bishop, 2011*). On this topic, only one meta-analysis study has explored the effects of CCT with a negligible effect on RSA ($p = 0.156$; ES $= 0.32$) in male soccer players (*Thapa et al., 2022*). However, some studies did not include control groups, so the efficacy is not adequately addressed. No previous studies have systematically explored the effects of ST and CCT on RSA in different team sports. Therefore, this systematic review aims to analyze the existing evidence on (1) The effects of ST on the RSA, (2) The effects of CCT on the RSA, and (3) The effects of ST and CCT on fatigue in team sports players. We hypothesized that ST and CCT would effectively improve the initial sprint performance (*i.e.,* best time) but not recovery between sprints (*i.e.,* fatigue index or percentage decrement).

## MATERIALS & METHODS

### Search strategy

This study is a systematic literature review, following the Preferred Reported Items for Systematic Reviews and Meta-Analysis (PRISMA) statements (*Liberati et al., 2009*), in which original, peer-reviewed articles published in English without date restrictions were considered. The PubMed, Web of Science (Core Collection), and Scopus databases were considered until May 26, 2024.

### Eligibility criteria

Inclusion and exclusion criteria were structured through a PICOS approach (participants, intervention, comparators, outcomes, and study design) and are presented in detail in Table 1.

**Table 1  Inclusion and exclusion criteria following the PICOS approach.**

| Category | Inclusion criteria | Exclusion criteria |
|---|---|---|
| **Population** | Male and female team sports players (field- or court-based invasion sports), without previous injuries or other health problems, of amateur or professional category, without age restriction | Non-team sports (*e.g.*, individuals, racquet or combat sports), or water-based team sports. Team sports players with health problems and injured |
| **Intervention** | Lower extremity strength training program, defined as the ability to exert force over some type of external object or resistance (*e.g.*, traditional strength or power training) (*Kraemer & Ratamess, 2004*), lower extremity complex/contrast training program, defined as a combination of heavy load strength exercise (slower speed) followed by low load plyometric/power exercise (faster speed), set by set fashion within the same session (*Cormier et al., 2022*) | Programs that included other motor gestures to strength or complex/contrast training of the lower extremity (*e.g.*, circuit training). Program that used only body weight (*e.g.*, plyometric or Nordic hamstring exercises) |
| **Comparator** | Active control group (CG) that only performed regular training in their respective team sport. It could include low-intensity exercise (*e.g.*, injury prevention) | Absence of an active control group |
| **Outcomes** | At least one measure of physical performance (running RSA test) before and after the intervention period. Also, RSA test was performed over the ground on a flat surface at maximal intensity, with a mean work duration of $\leq 10$ s or $\leq 80$ m in distance, a recovery duration of $\leq 60$ s and $\geq 2$ repetitions performed in total. Single set and multi-set repeated-sprints (*Thurlow et al., 2023*) | Lack of physical performance measures before and after the intervention period. RSA test was performed on a treadmill, cycle ergometers or another implement. RSA test was performed at submaximal intensity, with a work duration of $> 10$ s or $> 80$ m, a recovery duration of $> 60$ s, and only a single sprint repetition |
| **Study design** | Controlled and/or parallel trials | Non-controlled trials |

## Selection of articles

Search criteria were performed using the following descriptors: "ballistic training", "resistance training" [Mesh], "strength training" [Mesh], "complex training", "contrast training", "weight training", "repeated change of direction", "repeated sprint ability", "repeated sprint exercise". The search equation in PubMed, Web of Science (Core Collection), and Scopus databases originated with the Boolean operators "AND" and "OR". Table 2 shows in detail the search strategies used in the three databases.

In addition, manual searches were performed by consulting grey literature, and reference lists were analyzed from selected articles in search of possible studies.

## Extraction and evaluation process

Articles were selected by title and abstract; then, the full text was reviewed to finally apply the inclusion criteria described in Table 1. Duplicate articles were removed manually using Zotero (version 6.0.37) reference management software, and the full-text articles excluded were recorded in a Microsoft Excel table with the causes for their exclusion. Two authors conducted the search independently (AO-RY); if necessary, a third reviewer acted as judge (MMA) to resolve potential discrepancies between the two authors.

## Data extraction

Data extraction was performed by the lead author using a standardized form created in Microsoft Excel. The repeated sprint ability was considered the primary variable, measured

**Table 2  Search strategies used in each database.**

| Search equation | Filters | Databases | Results | Date |
|---|---|---|---|---|
| (("ballistic training"[Title/Abstract] OR "resistance training"[Title/Abstract] OR "strength training"[Title/Abstract] OR "complex training"[Title/Abstract] OR "contrast training"[Title/Abstract] OR "weight training"[Title/Abstract]) AND "repeated change of direction "[Title/Abstract]) OR "repeated sprint ability"[Title/Abstract] OR "repeated sprint exercise"[Title/Abstract] | Title/Abstract | PubMed | 664<br>678<br>723 | 14-05-2023<br>27-08-2023<br>26-05-2024 |
| (ballistic training OR resistance training OR strength training OR complex training OR contrast training OR weight training) AND (repeated change of direction OR repeated sprint ability OR repeated sprint exercise) | Topic: Searches title, abstract, author keywords, and Keywords Plus | Web of Science (Core Collection) | 812<br>832<br>893 | 14-05-2023<br>27-08-2023<br>26-05-2024 |
| ("ballistic training" OR "resistance training" OR "strength training" OR "complex training" OR "contrast training" OR "weight training" AND "repeated change of direction" OR "repeated sprint ability" OR " repeated sprint exercise") | All fields/Article | Scopus | 1,477<br>1,568<br>1,760 | 14-05-2023<br>27-08-2023<br>26-05-2024 |

with different RSA tests recorded in execution time in seconds as mean $\pm$ standard deviation (SD). From the extracted data, means, SD, first author's last name, year of publication of the studies, sport, competition season, training intervention characteristics (frequency (days/week), duration (weeks), total sessions (frequency/duration), general characteristics of training (protocol), pause between sets and exercises (seconds), rest between sessions (hours), intervention time (minutes)) and descriptive characteristics of team sports players (number of subjects per group (size), chronological age (years), height (cm), body weight (kg), sex (M/F), performance level and training experience (years of training)) were recorded in a results table.

## Study and assessment of the risk of bias

The Physiotherapy Evidence Database (PEDro) scale was used to assess the risk of bias and methodological quality of the included studies (*Moseley et al., 2002*). This scale was developed to quickly identify trials that tended to be internally valid and to have enough statistical information to guide decision-making. The PEDro scale is designed with eleven items, all but the first (external validity) awarding one point if present, so the final score should be between 0 and 10 points. According to the predetermined cut-off points, studies are classified as high (eight to ten), medium (four to seven), or low quality (less than four). (Table 3). The lead author assessed the risk of bias for all studies. Also, p and ES values were collected (if available) for each selected article. The percent of change ($\Delta\%$) was calculated (if not available) of each article to evaluate the magnitude of the effects using the following equation: $\Delta\% = (\text{Mpost} - \text{Mpre}/\text{Mpre}) \times 100$, where Mpost represents the mean value after intervention and Mpre the baseline mean value.

**Table 3  Risk of bias in studies.**

| Author | Criteria | | | | | | | | | | | | Study quality |
|---|---|---|---|---|---|---|---|---|---|---|---|---|---|
| | 1 | 2 | 3 | 4 | 5 | 6 | 7 | 8 | 9 | 10 | 11 | Total | |
| *Chatzinikolaou et al. (2018)* | 1 | 1 | 0 | 1 | 0 | 0 | 0 | 1 | 1 | 1 | 1 | 6 | Moderate |
| *Durán-Custodio et al. (2023)* | 1 | 1 | 0 | 1 | 0 | 0 | 0 | 0 | 1 | 1 | 1 | 5 | Moderate |
| *Gonzalo-Skok et al. (2016)* | 1 | 1 | 0 | 1 | 0 | 0 | 0 | 1 | 1 | 1 | 1 | 6 | Moderate |
| *Hammami et al. (2019)* | 1 | 1 | 0 | 1 | 0 | 0 | 0 | 1 | 1 | 1 | 1 | 6 | Moderate |
| *Hammami et al. (2018)* | 1 | 0 | 0 | 1 | 0 | 0 | 0 | 1 | 1 | 1 | 1 | 5 | Moderate |
| *Hammami et al. (2017b)* | 1 | 1 | 0 | 1 | 0 | 0 | 0 | 1 | 1 | 1 | 1 | 6 | Moderate |
| *Hammami et al. (2017a)* | 1 | 1 | 0 | 1 | 0 | 0 | 0 | 1 | 1 | 1 | 1 | 6 | Moderate |
| *Hermassi et al. (2017)* | 1 | 1 | 0 | 1 | 0 | 0 | 0 | 1 | 0 | 1 | 1 | 5 | Moderate |
| *Torres-Torrelo, Rodríguez-Rosell & González-Badillo (2017)* | 1 | 1 | 0 | 1 | 0 | 0 | 0 | 1 | 1 | 1 | 1 | 6 | Moderate |
| *Torres-Torrelo et al. (2018)* | 1 | 1 | 0 | 1 | 0 | 0 | 0 | 1 | 1 | 1 | 1 | 6 | Moderate |

**Notes.**

Note: **1.** eligibility criteria were specified **2.** subjects were randomly allocated to groups (in a crossover study, subjects were randomly allocated an order in which treatments were received) **3.** allocation was concealed **4.** the groups were similar at baseline regarding the most important prognostic indicators **5.** there was blinding of all subjects **6.** there was blinding of all therapists who administered the therapy **7.** there was blinding of all assessors who measured at least one key outcome **8.** measures of at least one key outcome were obtained from more than 85% of the subjects initially allocated to groups **9.** all subjects for whom outcome measures were available received the treatment or control condition as allocated or, where this was not the case, data for at least one key outcome was analysed by ''intention to treat'' **10.** the results of between-group statistical comparisons are reported for at least one key outcome **11.** the study provides both point measures and measures of variability for at least one key outcome.
A detailed explanation for each PEDro scale item can be accessed at the study by *Moseley et al. (2002)*.

## RESULTS

### Selection of the studies

Through the realization of the literature search in the selected databases, a total of 723 studies were identified in PubMed, 893 in Web of Science, and 1,760 from Scopus. After removing duplicates, meta-analyses, systematic reviews, and abstracts, a total of 2,807 results of full texts were available. After reading the titles and abstracts, 39 relevant articles remained, and 2,768 were excluded for not presenting the eligibility criteria. The full texts of the 39 articles were read using the inclusion and exclusion criteria specified for the eligibility of the studies. Finally, ten articles were selected (Fig. 1).

The methodological quality of the 10 studies included in this systematic review was quantified through the PEDro scale, yielding moderate quality in the 10 studies (five to six points). None of the studies reported whether the allocation was concealed (criterion 3) (Table 3).

### Study characteristics

Among the main characteristics of ten studies (*Chatzinikolaou et al., 2018*; *Durán-Custodio et al., 2023*; *Gonzalo-Skok et al., 2016*; *Hammami et al., 2019*; *Hammami et al., 2018*; *Hammami et al., 2017b*; *Hammami et al., 2017a*; *Hermassi et al., 2017*; *Torres-Torrelo, Rodríguez-Rosell & González-Badillo, 2017*; *Torres-Torrelo et al., 2018*), these are composed of 10 active control groups (continued with their regular training in their respective sport) and 11 experimental groups (replaced a part of the training of their respective sport by ST or CCT), of nine studies, included only one experimental group. Only one study experimented with more than one group (*Hammami et al., 2017a*). All studies were
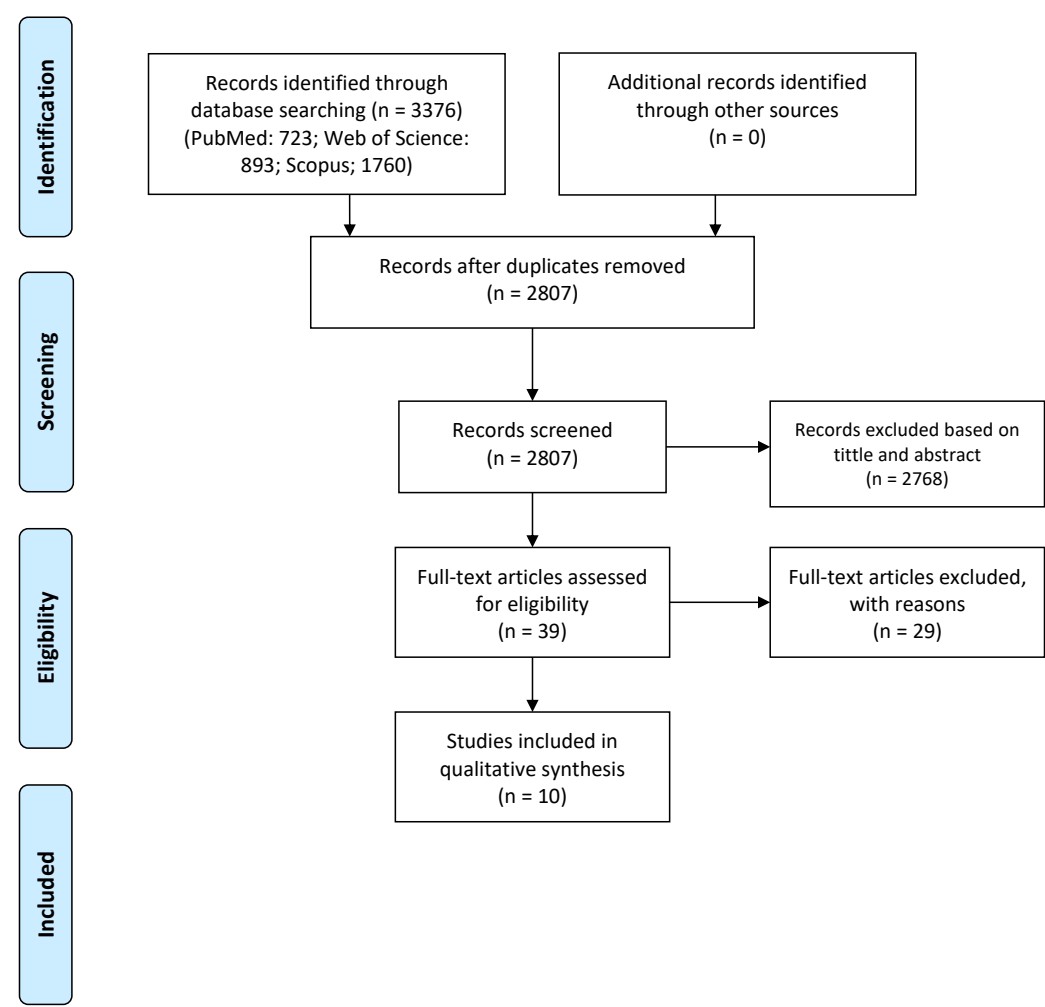

**Figure 1** PRISMA flow diagram.

randomized controlled trials except for one study (*Hammami et al., 2018*), which was not randomized. A total of nine studies included only men, and one included only women (*Hammami et al., 2019*). The age ranged from 14.1 ± 0.6 (*Chatzinikolaou et al., 2018*) to 25.8 ± 2.0 years (*Durán-Custodio et al., 2023*).

## Training and strength measurements

Regarding the types of training, four studies included traditional ST with free-weight using halfback squat (*Chatzinikolaou et al., 2018*; *Hammami et al., 2018*; *Hammami et al., 2017a*; *Hermassi et al., 2017*), three included CCT also with free-weight (*Chatzinikolaou et al., 2018*; *Hammami et al., 2017b*; *Hammami et al., 2017a*) but one (*Hammami et al., 2019*) with free-weight and machine, two studies included ST with free-weight using full squat with a linear velocity transducer (T-Force System, Ergotech, Murcia, Spain) to record bar velocity (*Torres-Torrelo, Rodríguez-Rosell & González-Badillo, 2017*; *Torres-Torrelo et al.,*

*2018*), one included ST with machine (Technogym™ and Precor™, Madrid, Spain) (*Durán-Custodio et al., 2023*) and another study included optimal power output load training with leg press (*Gonzalo-Skok et al., 2016*). Regarding movement velocity, five studies indicated "maximal voluntary effort" (*Hammami et al., 2019*; *Hammami et al., 2018*; *Hammami et al., 2017b*; *Hammami et al., 2017a*; *Hermassi et al., 2017*), three studies indicated "maximal concentric repetition velocity" (*Gonzalo-Skok et al., 2016*; *Torres-Torrelo, Rodríguez-Rosell & González-Badillo, 2017*; *Torres-Torrelo et al., 2018*), while two studies did not report it. All the studies included exercises in dynamic contractions (isotonic), multi-joint, and bilateral except the studies by *Chatzinikolaou et al. (2018)* and *Durán-Custodio et al. (2023)*, which included some unilateral exercises, besides the study by *Hammami et al. (2019)*, which included some exercises in static contractions (isometric) and single-joint.

### RSA testing
Regarding RSA testing, six studies included repeated shuttle sprint ability test $(6 \times 15 - 20 + 15 - 20 \text{ m}; 20 \text{ s passive recovery}; 180° \text{ turn})$ (*Gonzalo-Skok et al., 2016*; *Hammami et al., 2019*; *Hammami et al., 2018*; *Hammami et al., 2017b*; *Hammami et al., 2017a*; *Hermassi et al., 2017*) and four studies included straight-line repeated sprint ability Test $(5 - 9 \times 20 - 30 \text{ m}; 25 \text{ s active recovery})$ (*Chatzinikolaou et al., 2018*; *Durán-Custodio et al., 2023*; *Torres-Torrelo, Rodríguez-Rosell & González-Badillo, 2017*; *Torres-Torrelo et al., 2018*).

Also, the measures found in the RSA tests were as follows: the best time (the fastest of all sprints), the mean time (the average time of all sprints), the total time (the sum of all sprints), fatigue index (data not provided in the studies) and the percentage decrement score calculated as follows $(100 \times (\text{total sprint time} \div \text{ideal sprint time})) - 100$, where total sprint time = sum of sprint times from all sprints. Ideal sprint time = the number of sprints × fastest sprint time.

### Main results of strength and complex training (EG) on RSA
#### Effects of strength training on best time
Five studies provided data for best time. *Gonzalo-Skok et al. (2016)*, using maximal power training, found that for elite male basketball players, no substantial improvements were evident within-group (small ES; 0.24) or compared to CG (trivial ES; 0.2). Similar results were found by *Hammami et al. (2018)*; *Hammami et al. (2017a)*, both using traditional ST, where junior male soccer players had no significant changes compared to CG (all trivial ES; <0.2). Instead, *Hermassi et al. (2017)*, using traditional ST, found that elite male handball players had significant changes compared to CG (large ES [$\eta^2$]; 0.246). Also, *Torres-Torrelo et al. (2018)*, using ST with a linear velocity transducer, found that elite male futsal players had significant improvement within-group in best 10 m sprint time (moderate ES; −0.62) and best 20 m sprint time (small ES; −0.46) but this difference was not statistically different from the CG ($p > 0.05$, small ES; 0.50, 0.53).

#### Effects of strength training on mean time
Six studies provided data for mean time. *Gonzalo-Skok et al. (2016)* showed substantial improvements within-group (small ES; 0.49) with "likely" improvements compared to

CG. Instead, *Hammami et al. (2018)*; *Hammami et al. (2017a)* had no significant changes compared to CG (all trivial ES; <0.2). In contrast, *Hermassi et al. (2017)* had significant changes compared to CG (large ES ($\eta^2$); 0.183). Instead, *Torres-Torrelo, Rodríguez-Rosell & González-Badillo (2017)*, using ST with a linear velocity transducer, found that elite male futsal players had no significant changes within-group (trivial ES; −0.10) neither compared to CG ($p > 0.05$). Neither *Torres-Torrelo et al. (2018)* had any significant changes within-group in mean 10 m sprint time (trivial ES; −0.05) and mean 20 m sprint time (trivial ES; −0.11) neither compared to CG ($p > 0.05$, trivial ES; −0.06, 0.20).

### Effects of strength training on fatigue index or percentage decrement

Seven studies provided data for fatigue index or percentage (%) decrement. *Chatzinikolaou et al. (2018)*, using traditional ST and CCT in different sessions, had no significant intragroup changes ($p > 0.05$) in fatigue index in junior elite male soccer players, which neither were significant compared to the control condition. *Gonzalo-Skok et al. (2016)* showed "unclear" improvements within-group (small ES; 0.54) and compared to CG (small ES) in % decrement. *Hammami et al. (2018)*; *Hammami et al. (2017a)* had no significant effects compared to CG (all trivial ES; <0.2) in % decrement and fatigue index, respectively. Also, *Torres-Torrelo, Rodríguez-Rosell & González-Badillo (2017)* had no significant changes within the group in % decrement (small ES; 0.41) or compared to CG ($p > 0.05$). Instead, *Torres-Torrelo et al. (2018)* had significant changes within the group in % decrement 10 m (small ES; 0.46), but not % decrement 20 m (small ES; 0.44), neither compared to CG ($p > 0.05$, small ES; 0.42, 0.41) and *Durán-Custodio et al. (2023)*, using ST, found that semi-professional male soccer players had significant changes within the group (large ES; 1.25) in decrement (s) but not compared to CG ($p = 0.141$).

### Effects of strength training on total time

Four studies provided data for total time. *Hammami et al. (2018)*; *Hammami et al. (2017a)* had no significant changes compared to CG (all trivial ES; <0.2). Also, *Hermassi et al. (2017)* had no significant changes compared to CG (moderate ES ($\eta^2$); 0.078). In contrast, *Durán-Custodio et al. (2023)* had significant changes within the group (large ES; 5.34) and compared to CG ($p < 0.001$).

### Effects of complex/contrast training on best time, mean time, fatigue index, and total time

Three studies provided data for best time, mean time, fatigue index, and total time. *Hammami et al. (2019)*, using CCT, found that elite female handball players had significant changes compared to CG in all outcomes (all trivial ES; 0.08), except in the fatigue index (trivial ES; 0.01). Instead, the studies by *Hammami et al. (2017b)*; *Hammami et al. (2017a)*, both using CCT, found that junior male soccer players had no significant changes compared to CG in all outcomes (all trivial ES; <0.2).

### Effects of active control groups on RSA

The active control groups in the different studies continued their regular training in their respective team sport without performing ST or CCT and showed no significant intra-group changes in any measure. The descriptions of the selected studies, including the

**Table 4  Subjects' characteristics.**

| Author | Sport | Group | No of subjects (size) | Age (years) (mean ± SD) | Height (cm) (mean ± SD) | Weight (kg) (mean ± SD) | Sex (M/F) | Training experience (years) (mean ± SD) | Performance level | Season |
|---|---|---|---|---|---|---|---|---|---|---|
| *Chatzinikolaou et al. (2018)* | Soccer | STG | 12 | 14.3 ± 0.7 | 179 ± 0.1 | 69.13 ± 5.0 | M | ≥6/no experience in strength | Junior Elite | Off-season |
|  |  | CG | 10 | 14.1 ± 0.6 | 178 ± 0.1 | 72.24 ± 8.1 |  |  |  |  |
| *Durán-Custodio et al. (2023)* | Soccer | STG | 10 | 24.5 ± 3.1 | 176.6 ± 4.4 | 72.8 ± 6.6 | M | with experience in strength | Semi-professional | In-season |
|  |  | CG | 10 | 25.8 ± 2.0 | 178.3 ± 3.6 | 74.7 ± 5.7 |  |  |  |  |
| *Gonzalo-Skok et al. (2016)* | Basketball | RMPG | 11 | 16.2 ± 1.2 | 190.0 ± 10.0 | 82.9 ± 10.1 | M | 7 years/ with experience in strength | Elite | In-season |
|  |  | CG | 11 | 16.2 ± 1.2 | 190.0 ± 10.0 | 82.9 ± 10.1 |  |  |  |  |
| *Hammami et al. (2019)* | Handball | CCTG | 14 | 16.6 ± 0.3 | 163 ± 0.04 | 60.8 ± 4.7 | F | NR/with experience in strength | Elite | In-season |
|  |  | CG | 14 | 16.6 ± 0.3 | 164 ± 0.04 | 60.4 ± 4.3 |  |  |  |  |
| *Hammami et al. (2018)* | Soccer | STG | 19 | 16.2 ± 0.6 | 175 ± 0.03 | 58.1 ± 7.3 | M | NR/with experience in strength | Junior | In-season |
|  |  | CG | 12 | 15.8 ± 0.2 | 168 ± 0.05 | 58.2 ± 5.0 |  |  |  |  |
| *Hammami et al. (2017b)* | Soccer | CCTG | 17 | 16.0 ± 0.5 | 177 ± 5 | 58.9 ± 6.6 | M | NR/with experience in strength | Junior | In-season |
|  |  | CG | 12 | 16.8 ± 0.2 | 169 ± 5 | 58.4 ± 5.2 |  |  |  |  |
| *Hammami et al. (2017a)* | Soccer | STG | 16 | 16.2 ± 0.6 | 175 ± 0.03 | 58.0 ± 6.2 | M | NR/with experience in strength | Junior | In-season |
|  |  | CCTG | 16 | 16.0 ± 0.5 | 178 ± 0.05 | 59.3 ± 6.5 |  |  |  |  |
|  |  | CG | 12 | 16.8 ± 0.2 | 168 ± 0.05 | 58.1 ± 5.2 |  |  |  |  |
| *Hermassi et al. (2017)* | Handball | STG | 12 | 18.9 ± 0.2 | 195 ± 0.2 | 93.4 ± 10.2 | M | 9.2 ± 0.9 | Elite | In-season |
|  |  | CG | 10 | 18.9 ± 0.6 | 190 ± 0.3 | 91.2 ± 10.1 |  | 9.0 ± 0.6/with experience in strength |  |  |
| *Torres-Torrelo, Rodríguez-Rosell & González-Badillo (2017)* | Futsal | STG | 12 | 23.8 ± 2.4 | 177.2 ± 0.05 | 73.6 ± 7.0 | M | 10.0 ± 3.1 | Elite | In-season |
|  |  | CG | 10 | 24.7 ± 4.7 | 176.5 ± 0.06 | 75.9 ± 7.1 |  | 10.0 ± 3.3/no experience in strength |  |  |
| *Torres-Torrelo et al. (2018)* | Futsal | STG | 12 | 23.8 ± 2.4 | 177 ± 0.05 | 73.6 ± 7.0 | M | 10.0 ± 3.1 | Semi-professional | In-season |
|  |  | CG | 10 | 24.7 ± 4.7 | 177 ± 0.06 | 75.9 ± 7.1 |  | 10.0 ± 3.3/no experience in strength |  |  |

Notes.

Abbreviations: CG, control group; RMPG, repeated maximal power group; STG, strength training group; CCTG, complex contrast training group; M, male; F, female.

subjects' characteristics and the ST and CCT intervention, are shown in Tables 4 and 5. The main results of the RSA tests are in detail in Table 6.

## DISCUSSION

The present review aimed to delineate the evidence on the effects of ST and CCT on RSA in team sports players. The main findings were that ST and CCT performed ∼twice weekly, between six to twelve weeks, which led to some improvements in best, mean, and total sprint time. It is also worth highlighting that these interventions, in general, were ineffective in significantly improving the fatigue index or percentage decrement. In addition, these effects seemed to be present irrespective of playing level, age, or sex.

**Table 5  Strength and complex training intervention.**

| Author | F | D | TS | Training type | General training characteristics | Speed of movement | Rest between session (hours) | Time of intervention (minutes) |
|---|---|---|---|---|---|---|---|---|
| *Chatzinikolaou et al. (2018)* | 4 | 5 | 20 | **Strength** | 2 sessions (squat, Romanian deadlift (RDL), strides or step-ups, one-legged RDL) 60–70 to 80–90% of 1RM Sets: 2 Repetitions: 10–14 to 6–3 Pause: 60 s to 120 s | NR | NR | 30 |
| | | | | **Complex** | 2 sessions (combination of multi-joint Olympic-style lifts, plyometric, and speed work; *e.g.*, barbell cleans, kettlebell snatches, box jumps, speed/power drills) Sets: 3 Repetitions: 6–8 Pause: 90–120 s Linear Periodization [*] | NR | NR | 30 |
| *Durán-Custodio et al. (2023)* | 2 | 12 | 24 | **Strength** | Unilateral horizontal leg press, unilateral lateral leg press with 45° inclination of the supporting foot with respect to the surface, knee extension and knee flexion 85–95% of 1 RM Sets: 3 per each exercise Repetitions: 3–4 Pause: 3 min between sets Periodization progressive | NR | 48 | 35–40 |
| *Gonzalo-Skok et al. (2016)* | 2 | 6 | 12 | **Repeated maximal power** | Leg press exercise using the load that maximizes power output $120.3 \pm 22.1$ kg or $937.4 \pm 248.4$ watts. Weeks 1 to 3: 1 block Sets: 5 Repetitions: 5 Pause: 20 s passive recovery Weeks 4 to 6: 2 blocks Sets: 5 Repetitions: 5 Pause: 20 s between sets and 3 min between blocks, passive recovery Periodization progressive | Maximum speed concentric repetition. The eccentric phase was performed at a slower speed | 48 | 10–20 |

**Table 5** (*continued*)

| Author | F | D | TS | Training type | General training characteristics | Speed of movement | Rest between session (hours) | Time of intervention (minutes) |
|---|---|---|---|---|---|---|---|---|
| *Hammami et al. (2019)* | 2 | 10 | 20 | **Complex** | 4 times each series/session<br>Series 1: 6 repetitions half squat at 85% of 1 RM + 6 hurdle jumps to 40 cm + 10 m sprints<br>Series 2: 6 repetitions thigh press at 85% of 1 RM + 6 horizontal jumps + 10 m sprints<br>Series 3: half isometric squat of 8 s at 75% of 1RM + 6 jumps on 1 foot [3 on the right side and 3 on the left side] + 10 m sprints<br>Series 4: 6 repetitions of calf extension at 90% of 1RM + 6 hurdle jumps to 30 cm with legs extended + 10 m sprints<br>Pause: 1–2 min between sets<br>Linear Periodization [**] | Maximum verbal effort | NR | 40 |
| *Hammami et al. (2018)* | 2 | 8 | 16 | **Strength** | Halfback squat<br>Tuesday: ascending set 70–90% followed by a descending set 90–70% of 1RM<br>Thursday: upward set 70–90% of 1RM 3 sets x 8 reps at 70%<br>5 sets x 4 reps at 80%<br>4 sets x 3 reps at 85%<br>3 sets x 3 reps at 90%<br>Pause: NR<br>Linear Periodization [**] | Maximum verbal effort | NR | 30 |
| *Hammami et al. (2017b)* | 2 | 8 | 16 | **Contrast** | Weeks 1–4: halfback squat + 3 consecutive CMJs<br>Weeks 5–8: halfback squat + 1 CMJ + 15 m sprint<br>Tuesdays: ascending set 70–90% followed by a descending set 90–70% of 1RM<br>Thursday: upward set 70–90% of 1RM<br>3 sets x 8 reps at 70%<br>5 sets x 4 reps at 80%<br>4 sets x 3 reps at 85%<br>3 sets x 3 reps at 90%<br>Pause: NR<br>Linear Periodization [**] | Maximum verbal effort | NR | 45 |

| Author | F | D | TS | Training type | General training characteristics | Speed of movement | Rest between session (hours) | Time of intervention (minutes) |
|---|---|---|---|---|---|---|---|---|
| *Hammami et al. (2017a)* | 2 | 8 | 16 | **Strength** | Halfback squat<br>Tuesday: ascending set 70–90% followed by a descending set 90–70% of 1RM<br>Thursday: upward set 70–90% of 1RM<br>3 sets x 8 reps at 70%<br>5 sets x 4 reps at 80%<br>4 sets x 3 reps at 85%<br>3 sets x 3 reps at 90%<br>Pause: NR | Maximum verbal effort | NR | 45 |
| | | | | **Contrast** | Weeks 1–4: halfback squat + 3 consecutive CMJs<br>Weeks 5–8: halfback squat + 1 CMJ + 15 m sprint<br>Tuesdays: ascending set 70–90% followed by a descending set 90–70% of 1RM<br>Thursday: upward set 70–90% of 1RM<br>3 sets x 8 reps at 70%<br>5 sets x 4 reps at 80%<br>4 sets x 3 reps at 85%<br>3 sets x 3 reps at 90%<br>Pause: NR<br>Linear Periodization [**] | | | 45 |
| *Hermassi et al. (2017)* | 2 | 10 | 20 | **Strength** | Halfback squat<br>80 to 95% of 1 RM<br>Sets: 3-6<br>Repetitions: 3-1<br>Pause: 3-4 min between sets<br>Periodization Non-progressive | Maximum verbal effort | 48 | NR |
| *Torres-Torrelo, Rodríguez-Rosell & González-Badillo (2017)* | 2 | 6 | 12 | **Strength** | Full squat<br>∼1.20 m-s-1 (∼45% 1RM) to ∼1 m-s-1 (∼58% 1RM)<br>36.8 ± 8.0 kg to 49.6 ± 8.3 kg<br>Sets: 2-3<br>Repetitions: 6-4<br>Pause: 3 min between sets<br>Linear Periodization | Maximum speed concentric repetition | 48–72 | 20–25 |
| *Torres et al. (2018)* | 2 | 6 | 12 | **Strength** | Full squat<br>∼1.20 m - s-1 (∼45% 1RM) to ∼1 m - s-1 (∼60% 1RM)<br>36.8 ± 8.0 kg to 49.6 ± 8.3 kg<br>Sets: 2–3<br>Repetitions: 6-4<br>Pause: 3 min between sets<br>Linear Periodization | Maximum speed concentric repetition | 48–72 | 20–25 |

**Notes.**

Abbreviations: F, frequency (days/week); D, duration (weeks); TS, total sessions; 1 RM, repetition maximum; CMJ, countermovement jump; NR, not reported.

[*]The 1 RM value was reassessed every two weeks, and the strength loadings were adjusted accordingly.

[**]The 1 RM value was reassessed at the fourth week, and the strength loadings were adjusted accordingly.
**Table 6 Main results of the RSA tests.**

| Author | Physical test | Measure | Group | Pre-test (mean ± SD) | Post-test (Seconds) (mean ± SD) | Within-group changes | Within-group p-value, effect size (ES) | % Δ | Group x Time interaction p-value, effect size (ES) | Between-group Cohen's ES |
|---|---|---|---|---|---|---|---|---|---|---|
| *Chatzinikolaou et al. (2018)* | Repeated sprint ability Test RSA (s) (5 x 30 m; 25 s active rest) | Fatigue index (%) | STG<br>CG | N/A | N/A | →<br>→ | >0.05<br>>0.05 | N/A | >0.05 | N/A |
| *Durán-Custodio et al. (2023)* | Repeated sprint ability Test RSA (s) (5 x 30 m; 25 s active rest) | Total Time (s) | STG<br>CG | 22.34 ± 0.70<br>21.85 ± 0.43 | 21.50 ± 0.66<br>21.71 ± 0.45 | ↑<br>↑ | <0.001 (5.34)<br>0.010 (1.02) | −3.92<br>−0.64 | <0.001 (F)87.34 | |
| | | Decrement (%) | STG<br>CG | 4.15 ± 1.24<br>3.11 ± 0.95 | 3.10 ± 0.76<br>3.04 ± 0.68 | ↑<br>→ | 0.003 (1.25)<br>0.775 (0.09) | −33.94<br>−2.33 | 0.141 (F) 2.38 | |
| *Gonzalo-Skok et al. (2016)* | Repeated shuttle sprint ability Test RSSA (s) (6 x 20 + 20 m; 20 s passive rest) | Fastest Time (s) | RMPG<br>CG | 7.16 ± 0.23<br>7.17 ± 0.24 | 7.10 ± 0.18<br>7.17 ± 0.24 | →<br>→ | (0.24)<br>(−0.01) | −0.8<br>0.0 | Unclear | Small |
| | | Mean Time (s) | RMPG<br>CG | 7.52 ± 0.23<br>7.50 ± 0.24 | 7.40 ± 0.23<br>7.51 ± 0.22 | ↑<br>→ | (0.49)<br>(−0.03) | −1.6<br>0.1 | Likely [*] | Small |
| | | Slowest Time (s) | RMPG<br>CG | 7.86 ± 0.29<br>7.76 ± 0.27 | 7.67 ± 0.29<br>7.79 ± 0.22 | ↑<br>→ | (0.61)<br>(−0.13) | −2.4<br>0.5 | Likely [*] | Small |
| | | Percentage of decrement (%) | RMPG<br>CG | 5.1 ± 1.8<br>4.6 ± 1.8 | 4.3 ± 1.7<br>4.7 ± 2.2 | →<br>→ | (0.54)<br>(−0.04) | −19.0<br>1.9 | Unclear | Small |
| *Hammami et al. (2019)* | Repeated shuttle sprint ability Test RSSA (s) (6 x 20 + 20 m; 20 s passive rest) | Fastest Time (s) | CCTG<br>CG | 7.13 (0.32)<br>7.21 (0.13) | 7.01 (0.32)<br>7.16 (0.13) | ↑<br>→ | <0.05<br>>0.05 | −1.7<br>−0.7 | .050 (0.08) [*] | Trivial |
| | | Mean Time (s) | CCTG<br>CG | 7.39 (0.33)<br>7.44 (0.15) | 7.27 (0.33)<br>7.40 (0.15) | ↑<br>→ | <0.05<br>>0.05 | −1.6<br>−0.5 | .051 (0.08) [*] | Trivial |
| | | Fatigue index (%) | CCTG<br>CG | 3.71 (1.35)<br>3.11 (1.69) | 3.77 (1.38)<br>3.13 (1.70) | →<br>→ | >0.05<br>>0.05 | 1.6<br>0.6 | .527 (0.01) | Trivial |
| | | Total Time (s) | CCTG<br>CG | 44.4 (2.0)<br>44.6 (0.9) | 43.6 (2.0)<br>44.4 (0.90) | ↑<br>→ | <0.05<br>>0.05 | −1.8<br>−0.4 | .051 (0.08) [*] | Trivial |
| *Hammami et al. (2018)* | Repeated shuttle sprint ability Test RSSA (s) (6 x 20 + 20 m; 20 s passive rest) | Fastest Time (s) | STG<br>CG | 7.40 ± 0.11<br>7.37 ± 0.27 | 7.23 ± 0.07<br>7.28 ± 0.23 | →<br>→ | >0.05<br>>0.05 | −2.3<br>−1.2 | 0.380 (0.013) | Trivial |
| | | Mean Time (s) | STG<br>CG | 7.40 ± 0.11<br>7.37 ± 0.27 | 7.23 ± 0.07<br>7.28 ± 0.23 | →<br>→ | >0.05<br>>0.05 | −2.3<br>−1.2 | 0.958 (0.000) | Trivial |
| | | RSSA decrement (s) | STG<br>CG | 4.02 ± 1.42<br>3.41 ± 1.66 | 3.57 ± 1.39<br>3.27 ± 1.25 | →<br>→ | >0.05<br>>0.05 | −11.2<br>−4.1 | 0.132 (0.039) | Trivial |
| | | Total Time (s) | STG<br>CG | 46.2 ± 0.9<br>45.2 ± 2.8 | 45.0 ± 0.5<br>45.2 ± 1.7 | →<br>→ | >0.05<br>>0.05 | −2.6<br>0 | 0.113 (0.043) | Trivial |
| *Hammami et al. (2017b)* | Repeated sprint ability Test RSA (s) (6 x 20 m; 20 s active rest) | Fastest Time (s) | CCTG<br>CG | 7.39 ± 0.14<br>7.41 ± 0.35 | 7.23 ± 0.10<br>7.33 ± 0.35 | →<br>→ | >0.05<br>>0.05 | −2.2<br>−1.1 | 0.511 (0.007) | Trivial |
| | | Mean Time (s) | CCTG<br>CG | 7.64 ± 0.18<br>7.68 ± 0.38 | 7.45 ± 0.13<br>7.59 ± 0.39 | →<br>→ | >0.05<br>>0.05 | −2.5<br>−1.2 | 0.496 (0.008) | Trivial |

**Table 6** (*continued*)

| Author | Physical test | Measure | Group | Pre-test (mean ± SD) | Post-test (Seconds) (mean ± SD) | Within-group changes | Within-group p-value, effect size (ES) | % Δ | Group x Time interaction p-value, effect size (ES) | Between-group Cohen's ES |
|---|---|---|---|---|---|---|---|---|---|---|
| | | Fatigue index (%) | CCTG | 3.44 ± 1.14 | 3.29 ± 1.04 | → | >0.05 | −4.4 | 0.243 (0.023) | Trivial |
| | | | CG | 2.26 ± 4.67 | 3.52 ± 1.46 | → | >0.05 | 55.8 | | |
| | | Total Time (s) | CCTG | 45.84 ± 1.11 | 44.80 ± 0.73 | → | >0.05 | −2.3 | 0.257 (0.022) | Trivial |
| | | | CG | 45.45 ± 3.22 | 45.53 ± 2.34 | → | >0.05 | 0.2 | | |
| *Hammami et al. (2017a)* | Repeated shuttle sprint ability Test RSSA (s) (6 x 20 + 20 m; 20 s passive rest) | Fastest Time (s) | CCTG | 7.41 ± 0.15 | 7.23 ± 0.09 [#] | → | >0.05 | −2.4 | 0.656 (0.010) | Trivial |
| | | | STG | 7.42 ± 0.13 | 7.24 ± 0.08 | → | >0.05 | −2.4 | | |
| | | | CG | 7.44 ± 0.38 | 7.37 ± 0.36 | → | >0.05 | −0.9 | | |
| | | Mean Time (s) | CCTG | 7.64 ± 0.20 | 7.46 ± 0.13 [#] | → | >0.05 | −2.4 | 0.724 (0.008) | Trivial |
| | | | STG | 7.68 ± 0.13 | 7.60 ± 0.17 | → | >0.05 | −1.0 | | |
| | | | CG | 7.71 ± 0.39 | 7.62 ± 0.40 | → | >0.05 | −1.2 | | |
| | | Fatigue index (s) | CCTG | 3.16 ± 1.09 | 3.46 ± 1.02 [#] | → | >0.05 | 9.5 | 0.777 (0.006) | Trivial |
| | | | STG | 3.57 ± 1.08 | 3.66 ± 1.37 | → | >0.05 | 2.5 | | |
| | | | CG | 2.18 ± 4.67 | 3.39 ± 1.41 | → | >0.05 | 55.5 | | |
| | | Total Time (s) | CCTG | 45.85 ± 1.18 | 44.87 ± 0.72 [#] | → | >0.05 | −2.1 | 0.378 (0.023) | Trivial |
| | | | STG | 46.10 ± 0.79 | 45.00 ± 0.53 | → | >0.05 | −2.4 | | |
| | | | CG | 45.65 ± 3.30 | 45.69 ± 2.38 | → | >0.05 | 0.08 | | |
| *Hermassi et al. (2017)* | Repeated shuttle sprint ability Test RSSA (s) (6 x 15 + 15 m; 20 s passive rest) | Fastest Time (s) | STG | 6.74 ± 0.31 | 6.24 ± 0.12 | ↑ | <0.05 (2.33) | −7.4 | 0.019 (0.246) [*] | Large ($\eta^2$) |
| | | | CG | 6.76 ± 0.33 | 6.60 ± 0.13 | → | >0.05 (0.70) | −2.4 | | |
| | | Mean Time (s) | STG | 6.93 ± 0.31 | 6.51 ± 0.08 | ↑ | <0.05 (2.15) | −6.06 | 0.047(0.183) [*] | Large |
| | | | CG | 6.97 ± 0.33 | 6.81 ± 0.10 | → | >0.05 (0.74) | −2.3 | | |
| | | Total Time (s) | STG | 40.5 ± 3.39 | 39.1 ± 0.45 | → | >0.05 (0.73) | −3.5 | 0.208 (0.078) | Moderate |
| | | | CG | 40.4 ± 3.73 | 41.0 ± 1.38 | → | >0.05 (−0.24) | 1.5 | | |
| *Torres-Torrelo, Rodríguez-Rosell & González-Badillo (2017)* | Repeated sprint ability Test RSA (s) (9 x 20 m; 25 s active rest) | Mean Time (s) | STG | 3.17 ± 0.12 | 3.16 ± 0.12 | → | >0.05 (−0.10) | −0.4 | 0.025 [*] | Trivial |
| | | | CG | 3.19 ± 0.11 | 3.20 ± 0.11 | → | >0.05 (0.09) | 0.3 | | |
| | | Decrement (%) | STG | 3.60 ± 1.43 | 4.81 ± 2.62 | → | >0.05 (0.41) | 25.6 | >0.05 | Trivial |
| | | | CG | 5.41 ± 2.99 | 5.27 ± 3.07 | → | >0.05 (0.02) | 1.2 | | |
| *Torres-Torrelo et al. (2018)* | Repeated sprint ability Test RSA (s) (9 x 20 m; 25 s active rest) | Fastest Time 10 m (s) | STG | 1.74 ± 0.05 | 1.71 ± 0.08 | ↑ | <0.05 (−0.62) | −1.9 | >0.05 (0.50) | Small |
| | | | CG | 1.73 ± 0.06 | 1.73 ± 0.06 | → | >0.05 (−0.06) | −0.2 | | |
| | | Fastest Time 20 m (s) | STG | 3.06 ± 0.10 | 3.02 ± 0.10 | ↑ | <0.05 (−0.46) | −1.6 | <0.05 (0.53) [*] | Small |
| | | | CG | 3.03 ± 0.09 | 3.04 ± 0.08 | → | >0.05 (0.05) | 0.2 | | |
| | | Mean Time 10 m (s) | STG | 1.82 ± 0.07 | 1.82 ± 0.08 | → | >0.05 (−0.05) | 0.0 | >0.05(−0.06) | Trivial |
| | | | CG | 1.84 ± 0.06 | 1.83 ± 0.06 | → | >0.05 (−0.13) | −0.4 | | |
| | | Mean Time 20 m (s) | STG | 3.17 ± 0.12 | 3.16 ± 0.12 | → | >0.05 (−0.13) | −0.4 | <0.05 (0.20) [*] | Small |
| | | | CG | 3.19 ± 0.11 | 3.20 ± 0.11 | → | >0.05 (−0.11) | 0.3 | | |
| | | Decrement 10 m (%) | STG | 8.07 ± 3.88 | 10.49 ± 5.02 | ↑ | <0.05 (0.46) | 26.8 | >0.05 (0.42) | Small |
| | | | CG | 11.57 ± 6.10 | 11.00 ± 5.32 | → | >0.05 (−0.02) | −1.3 | | |
| | | Decrement 20m (%) | STG | 6.47 ± 2.50 | 8.05 ± 3.44 | → | >0.05 (0.44) | 20.5 | 0.05 (0.41) | Small |
| | | | CG | 9.59 ± 4.51 | 9.80 ± 6.02 | → | >0.05 (0.05) | 2.4 | | |

**Notes.**

Abbreviations: CG, control group; RMPG, repeated maximal power group; STG, strength training group; CCTG, complex contrast training group p <0.05; N/A, not available; ↑, presented significant change within-group; →, no significant change within-group.

[*]Significant "group x time" interaction.

[#]A main effect of time.

### Effects of strength training on the RSA

Four studies (*Durán-Custodio et al., 2023*; *Gonzalo-Skok et al., 2016*; *Hermassi et al., 2017*; *Torres-Torrelo et al., 2018*) presented significant changes in their EG in specific measures in the RSA tests (best time, mean time, and total time). These results are coincident with previous studies that only performed ST with similar exercises of the lower extremity (although with different % loads) and similar RSA tests in young male soccer players, professional male futsal players, and elite male soccer players, respectively (*Sanchez-Sanchez et al., 2022*; *Paz-Franco, Rey & Barcala-Furelos, 2017*; *Spineti et al., 2016*).

The improvements in initial sprint performance (*i.e.,* best time) after heavy ST and maximal power training may be attributed to specific neural adaptations (ability to activate all high threshold MUs with maximum discharge rate rapidly) and morphological adaptations (cross-sectional area, muscle architecture, and muscle–tendon stiffness) which are achieved by executing the concentric phase of lifting with the maximum intensity. Previous research indicated that both heavy ST and explosive ST are effective in improving RFD (*i.e.,* the ability of the neuromuscular system to express the greatest amount of force in the shortest possible time) (*Maffiuletti et al., 2016*; *Suchomel, Nimphius & Stone, 2016*). Therefore, an increase in the propulsion impulse and RFD could allow a greater force to be produced over a given time, resulting in greater acceleration and shorter ground contact times (*Maffiuletti et al., 2016*; *Suchomel, Nimphius & Stone, 2016*). Another possible explanation, according to *Seitz et al. (2014)*, is the increases in lower-body strength levels after ST, using a medium (60–84.9% 1 RM; ES $= -0.97$) and high ($>85\%$ 1 RM; ES $= -0.52$) intensity, transfer positively to sprint performance (higher peak GRF, impulse and RFD) ($r = -0.77$; $p = 0.0001$).

For example, strong relationships have been found between 1 RM strength of the half squat and 10 m ($r = 0.94$; $p < 0.001$) and 30 m sprint time ($r = 0.71$; $p < 0.01$), respectively, in elite male soccer players (*Wisløff et al., 2004*). Also, relationships between full squats using relative loads that maximize the mechanical power output have been found ($\sim60\%$ 1 RM; mean propulsive velocity of $\sim1.00$ m· s $-1$) and best time ($r = -0.76$; $p = 0.007$), mean time of the first 3 sprints ($r = -0.64$; $p = 0.004$) and mean time of 9 sprints ($r = -0.52$; $p = 0.04$) in under-19 soccer players (*López-Segovia et al., 2015*). Although this suggests that strength qualities are interrelated with sprint performance, this does not prove causality; however, the present results support these findings.

On the other hand, the improvements in the mean time (*Gonzalo-Skok et al., 2016*; *Hermassi et al., 2017*) and total time (*Durán-Custodio et al., 2023*) could be attributed to an increase in RFD and/or power through adaptations in the stiffness of the muscle–tendon unit, which result from more efficient stretch-shortening cycles (*i.e.,* improvement storage and reuse of elastic energy during the braking and propulsion phases) (*Li et al., 2019*). In particular, mean time is related to the initial sprint performance due to a perfect correlation ($r = 0.93$; $p = 0.000$) between the best time and mean time of the first 3 sprints, whereas this relationship decreases for the subsequent sprints (*López-Segovia et al., 2015*). In contrast, the lack of improvements in mean and total time in the other studies could be explained because despite maximal strength increases (1 RM) in six studies (Table S1), the importance of 1RM decreases when the number of sprints increases. Here, the aerobic

capacity could be more relevant for improving recovery between sprints and/or avoiding the drop performance in the subsequent sprints (*López-Segovia et al., 2015*). Therefore, other, more specific training methods are needed to improve RSA. Previous research found a positive relationship between 1 RM strength of the half squat and best time ($r = 0.811$; $p < 0.001$) and total time ($r = 0.625$; $p = 0.002$) (*Hermassi et al., 2019*), also in the mean time ($r = 0.50$; $p < 0.05$) and total time ($r = 0.78$; $p = 0.01$) in male handball players respectively (*Hermassi et al., 2014*). In other words, the subjects with greater 1 RM performed the worst results on the RSA test.

The main mechanical factor determining RSA performance is the technical ability to apply a large forward-oriented ground reaction force vector effectively (*i.e.,* ground reaction impulse in the forward direction of the movement) during the acceleration of each sprint instead of the amount (magnitude) of total force applied, minimizing the decrease in the stride frequency (*Morin et al., 2011*; *Brocherie et al., 2016*; *Jiménez-Reyes et al., 2019*). Exercises of high force to low speed with a horizontally directed force vector would be more specific for improving forward acceleration from a static start or reacceleration from each sprint (*Hicks et al., 2019*). Therefore, the lack of improvements in the best and mean time (*Gonzalo-Skok et al., 2016*; *Hammami et al., 2018*; *Hammami et al., 2017a*; *Torres-Torrelo, Rodríguez-Rosell & González-Badillo, 2017*) could be due in part to the lack of a specific stimulus (force vector theory) during the acceleration of each sprint (*Loturco et al., 2018*). However, *Junge et al. (2021)* did not find significant associations between hip thrust exercise and acceleration in 30 m ($r = 0.18$; $p = 0.36$). In contrast, *Loturco et al. (2018)* found a strong association ($r = 0.93$) between hip thrust with maximal acceleration (zero to 10 m) and jump squats with sprints above 40 m in both sprinters and high-level jumpers. However, studies that showed improvement in RSA used similar exercises of back squats in the vertical direction as those that did not show improvement. Therefore, the force vector theory is inconclusive because it has only been measured in single-sprint tests, not RSA tests. On the other hand, there is no relation between straight-line and shuttle sprint performance (*Wisløff et al., 2004*); we speculate that the lack of improvements in the studies (*Gonzalo-Skok et al., 2016*; *Hammami et al., 2018*; *Hammami et al., 2017a*) could be because the strength exercises do not accentuate the eccentric phase, since the straight-line sprint with one 180° change of direction requires the inclusion of a deceleration movement before a reacceleration.

### Effects of complex training on the RSA

Regarding the effects of CCT on RSA, one of three studies (*Hammami et al., 2019*) reported a trivial significant change in their EG in specific measures (best time, mean time, and total time) in the RSA test. This result coincides with the meta-analysis of *Thapa et al. (2022)*, where three studies on youth, junior, and professional male soccer players found a small non-significant effect ($p = 0.156$; ES $= 0.32$) in the RSA tests.

After the CCT, improvements in the RSA test may be specifically related to optimizing the entire spectrum of the force-velocity curve in the same session (*Thapa et al., 2022*). Heavy load training and low speed (*e.g.,* squat 90% of 1RM) increase the excitability of the MUs (*i.e.,* recruitment and synchronization of MU higher order/high threshold and the

potentiation of reflexes), creating optimal and favorable conditions for plyometric or power training (*e.g.,* CMJ) after maximal muscle contractile activity (*Freitas et al., 2017*; *Cormier et al., 2022*). These neural adaptations may ultimately lead to a higher RFD (*Maffiuletti et al., 2016*) and improved efficiency of the stretch-shortening cycle. However, the time course of the potentiation effect also depends on the time course of fatigue (the net balance determining if there is an increased or decreased output).

Previous studies compared CCT *vs.* ST. *Spineti et al. (2016)* observed that CCT induced a significant improvement in percentage decrement (moderate ES; $-0.91$) and mean time (moderate ES; $-0.85$) when compared with ST in under-20 elite male soccer players. *Hammami et al. (2017a)* observed that CCT presented more significant changes for the mean ($p \leq 0.001$) and total time ($p \leq 0.01$) compared to ST in junior male soccer players. This suggests that CCT could be an effective strategy to optimize and improve performance during the competitive season compared to ST alone. However, the methodology used in the present systematic review and the lack of studies do not allow a comparison of the results to conclude that CCT is superior to ST. The lack of improvements in the studies by *Hammami et al. (2017b)*; *Hammami et al. (2017a)*, could be due to the lack of a specific stimulus, as aerobic capacity is an important factor in recovery between sprints; CCT does not develop this quality. Also, it could be due to a nonspecific recovery post-CA (*i.e.,* the relation between the CA on subsequent exercises) (*Cormier et al., 2022*); however, none of the studies reported the pause intra-contrast rest to affirm it.

## Effects of strength and complex training on fatigue

Fatigue was assessed through two measures (fatigue index or percentage decrement); only one study (*Durán-Custodio et al., 2023*) presented a large significant change ($p = 0.003$; ES $= 1.25$), except for the study by *Hermassi et al. (2017)* that did not report this measure (does not allow us to know its meaning). However, this unique positive result does not coincide with other studies (*Paz-Franco, Rey & Barcala-Furelos, 2017*; *Spineti et al., 2016*). These results should be interpreted with caution because a better initial performance of the first sprint will commonly lead to a high fatigue index (an indicator of the loss of performance between the best and the worst sprint), given that there is a positive correlation with the decrease performance in the following sprints; therefore, the subjects with higher anaerobic power reserve could be impaired in subsequent bouts (*Girard, Mendez-Villanueva & Bishop, 2011*). Probably, this decrease in performance would be a combination of neural factors (*i.e.,* lower recruitment and discharge frequency fast twitch MUs), as well as alterations in intermuscular coordination patterns (agonist-antagonist ratio) (*Girard, Mendez-Villanueva & Bishop, 2011*) and muscular (*i.e.,* accumulation of hydrogen ions ($H^+$), inorganic phosphate and extracellular potassium secondary to decreased sodium-potassium pump ATPase activity), also limitations in the energy supply of phosphocreatine availability, anaerobic glycolysis and oxidative metabolism (*Girard, Mendez-Villanueva & Bishop, 2011*), that may alter the sprint mechanics, such as reductions in forward-oriented ground reaction force (total force), increase in the contact time (*i.e.,* decreased efficiency of the stretch-shortening cycle), increased vertical displacement of the center of mass during the braking phase and increased swing time, reductions in vertical

stiffness and stride frequency (*Morin et al., 2011*; *Brocherie et al., 2016*; *Jiménez-Reyes et al., 2019*). In summary, ST or CCT would influence the neuro-mechanical properties of initial sprint performance; however, other training methods targeting oxidative capacity, phosphocreatine recovery, and $H^+$ buffering would be the most suitable for improving recovery between sprints and/or fatigue resistance (*Bishop, Girard & Mendez-Villanueva, 2011*; *Gaitanos et al., 1993*). Nevertheless, *Edge et al. (2006)* and *Hill-Haas et al. (2007)* showed in females recreationally active in various team sports that high-repetition resistance training that includes a high metabolic load (because of short rest periods 20 s) improved not only single-sprint performance but also the performance of subsequent sprints (peak and mean power). Although those studies were evaluated on a cycloergometer and did not comply with the principle of specificity, high-repetition resistance training is an effective method to improve the $H^+$ regulation and avoid the drop performance in subsequent bouts.

Some limitations can be identified. Firstly, since allocation was concealed, the moderate quality of the studies could produce systematic biases in random allocation. Also, it is difficult to mask the application of treatments or programs. The small number of studies on the effectiveness of ST on RSA does not allow us to compare the results regarding the characteristics of the subjects and intervention of the training. Furthermore, the heterogeneity of the characteristics of the subjects and intervention are factors to consider, mainly because subjects with lower levels of strength and experience could have a more considerable margin of adaptations. Future studies should include exercises in all the spectrum of the force-velocity curve, including exercises in the horizontal direction, ballistic where higher RFDs can be achieved, and high repetition with short rest periods. Finally, the diversity of RSA tests and the lack of measurements in 5 studies do not precisely allow us to know the interventions' efficacy.

# CONCLUSIONS

The findings of this systematic review suggest that ST performed in a range of maximal power provides benefits in the best time and mean time and performed between 80 to 95% of 1 RM provides benefits in the best time, mean time, and total time of the RSA tests, respectively, in elite and semi-professional male players during the competitive season. CCT performed between 75 to 90% of 1 RM combined with jumps and sprints provides benefits in the best time, mean time, and total time of the RSA test in elite female players during the competitive season; however, only one study reported a trivial significant change, it would not be correct to report that it is effective.

The improvements in the best time, mean time, and total time may be due to neuro-mechanical factors that affect initial sprint performance rather than the capacity to recover between efforts.

## Practical applications

According to the findings of the articles reviewed in the present study, coaches should consider ST and CCT to improve RSA in team sports. Specifically, training maximal strength, RFD, and power using a combination of heavy/low load strength exercises while

executing the concentric phase with maximum intensity may improve the propulsion impulse over the ground during the acceleration or reacceleration of each single sprint. Including exercises specific in a horizontal direction should also be considered to minimize the reduction in forward-oriented GRF during repeated sprinting, which could lead to important changes in RSA.

### Funding
The authors received no funding for this work.

### Competing Interests
The authors declare there are no competing interests.

### Author Contributions
- Andrés Osses-Rivera conceived and designed the experiments, performed the experiments, analyzed the data, prepared figures and/or tables, authored or reviewed drafts of the article, and approved the final draft.
- Rodrigo Yáñez-Sepúlveda analyzed the data, prepared figures and/or tables, authored or reviewed drafts of the article, and approved the final draft.
- Sebastián Jannas-Vela analyzed the data, prepared figures and/or tables, authored or reviewed drafts of the article, and approved the final draft.
- Jeppe F. Vigh-Larsen analyzed the data, prepared figures and/or tables, authored or reviewed drafts of the article, and approved the final draft.
- Matías Monsalves-Álvarez conceived and designed the experiments, performed the experiments, analyzed the data, prepared figures and/or tables, authored or reviewed drafts of the article, and approved the final draft.

### Data Availability
This is a systematic review.

### Supplemental Information
Supplemental information for this article can be found online at http://dx.doi.org/10.7717/peerj.17756#supplemental-information.

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
