# Peer review of "Effects of strength training on repeated sprint ability in team sports players: a systematic review"

_PeerJ, doi:10.7717/peerj.17756_

## Round 0.1 · original submission · Major Revisions

Both authors are of the opinion that the authors would need to provide stronger justification for the review and include novel analysis that has not been previously conducted.

**Language Note:** The review process has identified that the English language must be improved. PeerJ can provide language editing services - please contact us at [email protected] for pricing (be sure to provide your manuscript number and title). Alternatively, you should make your own arrangements to improve the language quality and provide details in your response letter. – PeerJ Staff

Reviewer 1 ·

Basic reporting

General comments
The study investigates the effect of strength and complex training upon RSA and found only 9 articles. Furthermore, of the 9 articles it seems that it is from 4 different research groups, which is very limited for a systematic review. I should expect that there would be more studies on effect of strength and complex training upon repeated sprint performance in team sports.
There are several parts of the review that are very long and do not make a clear point. This makes the flow of the story not very good. Furthermore, there are many speculations /theories of how the different training programs help, while there are no direct results if these theories are correct for those findings. Too much general theories.
Why were only strength and complex training chosen and define clearly what both types of training are.
I suggest to start with the pedroscale in the results to indicate the quality of the studies first.
The results part is way too long and thereby difficult to follow. Rewrite it.
The writing of using the references is also weird sometimes. You may write Hammami et al (2019) found that ….
First part of the discussion is again results and not discussion. Moreover, the discussion is also very long and speculates a lot with using general theories to explain the findings. However, the theories are explained too much, which are not shown by numbers in the studies you refer to. Thereby it is difficult if the findings of the studies in the review follow the general theories.
Some tables are difficult to follow, especially table 3.

Specific comments
Lines 107- 119: what is the point of this part of the paragraph
Lines 130-138: point with this part.
Line 305 per week
Line 367 intergroup changes

Experimental design

Se comments above

Validity of the findings

See comments above

Reviewer 2 ·

Basic reporting

The review aimed to analyze the existing evidence on the effects of strength and complex training on repeated sprint ability (RSA) in team sports players. While this may potentially provide important information to readers, the current review lacks nolvelty as information provided does not differ much from previous studies. The authors would need to provide stronger justification for the review and include novel analysis that has not been previously conducted. I have provided my suggestion which may help to improve the review.

L 106 – “determined by the length and stride frequency” amend to “determined by the stride length and frequency”
L107 – consider amending to “elastic properties of musculotendinous unit/tissue”
L107 – add reference/s after flexibility
L107 – “stride frequencies are” amend to “stride frequency is”
L 116 – add reference/s after aerobic capacity.
L125 – “However, since initial sprint performance is associated with a more significant subsequent drop in performance, it is unclear how a potential strength-mediated improvement in sprint performance affects repeated sprint performance when several bouts are performed”
This statement came up too early, it breaks the flow of your write up. Would be better to move it to later paragraph. Instead, introduce complex training as one of the training methods first.
L128-138 – “The development of maximal strength, RFD, and 129 power at the neural level follows Hennemanís size…..” this portion is not necessary.
L140 – Briefly introduce different methods of strength training before focusing on complex training.
L143 – “post-activation potentiationî (PAP)” Please be update on the current consensus of terminology. It should be postactivation performance enhancement. And also be updated on the mechanisms of enhanced performance. Refer to Blazevich and Babault 2019 as you have referenced. https://doi.org/10.3389/fphys.2019.01359
L150 – You can include the following portion around here, “However, since initial sprint performance is associated with a more significant subsequent drop in performance, it is unclear how a potential strength-mediated improvement in sprint performance affects repeated sprint performance when several bouts are performed”
Basically highlighting all the limitations at the end to justify why the review was conducted.
L154 – “We hypothesized that these interventions would be effective in improving the initial sprint performance and mean times of each sprint due to a close relationship between both variables”
This hypothesis does not address the question highlighted in L125.
Also, the current justification of the review is centered on “more sports” as compared to Thapa et al. 2020 which focused on football athletes. This is not exactly a strong justification. The demands of repeated sprint does not change regardless of sport.

Discussion
- The authors mentioned a fair bit on RFD. However, no information on improvement in RFD was provided in the tables and result. This suggests that the interpretation is speculative. I suggest that the authors conduct a correlation analysis between strength and RSA improvement, and RFD and RSA improvement for the included studies if those data are available.
- Studies that showed improvement in RSA have used similar exercises for training as those that did not show improvement. So the mention on exercise usage does not really justify the lack of improvement due to the lack of movement specificity.
- The authors did not explicitly discuss about the difference in RSA adaptations between traditional strength training and complex training.
- In conclusion, the authors stated that strength training provides benefits in the best time and mean time, this was not explicitly highlighted and discuss in the discussion section.
- Back to the statement in L125, the discussion section did not address that question, which I feel that should be the main justification and novelty for this review.

Experimental design

As above

Validity of the findings

As above

Additional comments

As above

---

## Round 0.2 · Major Revisions

One of the earlier reviewers recommended the rejection of the manuscript. Hence, we engaged a third reviewer for an additional view (R3).

Please provide a point-by-point reply to this reviewer´s concerns.

Reviewer 1 ·

Basic reporting

I see that the authors have made some changes to the review. However, in my opinion does it still not hold an high enough standard and novelty for publication. Sorry

Experimental design

The design is ok, but still only using strength training and complex training as the only intervention methods is not enough.

Validity of the findings

The findings are limited due to the limitation of the two types of training interventions and thereby not giving a fully view over possible training methods that could enhance RSA performance. Furthermore, data is from a very limited group of researchers (Tunesia), which in my opinion is also a large limitation of this review. This would mean that in the rest of the world nobody does this type of intereventions and look at RSA?

Reviewer 2 ·

Basic reporting

The authors have addressed my queries, I have no further comments.

Experimental design

NA

Validity of the findings

NA

Additional comments

NA

·

Basic reporting

This systematic review was conducted to analyze the existing evidence on the effects of strength and complex training on repeated sprint ability (RSA) in team sports players compared to a control group. It is an interesting manuscript with a robust research design, however some sections need some major revisions.

Experimental design

Please consider the following point-by-point revisions in the attachment.

Validity of the findings

Please consider the following point-by-point revisions in the attachment.

---

## Round 0.3 · accepted · Accept

The 3rd reviewer also recommended the acceptation of this manuscript.

·

Basic reporting

After extensive peer review, all comments and suggestions were answered clearly and the manuscript is robust, well-supported and up-to-date research topics.

Experimental design

Nothing more to report.

Validity of the findings

Nothing more to report.

Additional comments

Nothing more to report.